

# Competition between frustration and spin dimensionality in the classical antiferromagnetic $n$-vector model with arbitrary $n$

## Nikolaos P. Konstantinidis⋆

Department of Mathematics and Natural Sciences, The American University of Iraq, Sulaimani, Kirkuk Main Road, Sulaymaniyah, Kurdistan Region, Iraq

⋆ npknpk1111@gmail.com

## Abstract

A new method to characterize the strength of magnetic frustration is proposed by calculating the minimum dimensionality of the absolute ground states of the classical nearest-neighbor antiferromagnetic $n$-vector model with arbitrary $n$. Platonic solids in three and four dimensions and Archimedean solids have lowest-energy configurations in a number of spin dimensions equal to their real-space dimensionality. Fullerene molecules and geodesic icosahedra can produce ground states in as many as five spin dimensions. Frustration is also characterized by the maximum value of the round-state energy when the exchange interactions are allowed to vary.

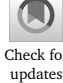

# 1  Introduction

Strongly-correlated electron interactions can very often be described by effective Hamiltonians involving only spin degrees of freedom [1, 2]. A Hamiltonian describing interactions between localized spins is the nearest-neighbor antiferromagnetic Heisenberg model, which has been considered on various lattices and clusters. A single spin is located at each vertex of the underlying structure and only nearest-neighbors interact. It is of special interest when the underlying structure's topology is frustrated, meaning that due to competing interactions not all nearest-neighbor spin pairs are simultaneously antiparallel in the ground state when the spins are classical [3–7].

## 1.1  Frustration

In the nearest-neighbor antiferromagnetic Ising model the spins interact only along a single direction in spin space, making it maximally anisotropic [8]. The dimensionality of the interactions can be increased to two and three in a controlled way, by allowing the spins to interact in the added spin dimensions with increasing strength. The corresponding models are the XY in two dimensions and the anisotropic Heisenberg, or XXZ, in three. When the interaction components become equal in all two or three three spin directions one gets respectively the XX model and the isotropic XXX Heisenberg model. Frustration manifests itself in all these cases but is more pronounced in the XXX limit, where magnetic anisotropy plays no role. Then any deviation from antiparallel nearest-neighbor classical spins in the zero-magnetic-field ground state and any magnetization or susceptibility discontinuities in an external field are solely due to the frustrated connectivity of the structure that hosts the spins.

In the case of bipartite structures, which lack frustration, nearest-neighbors point at antiparallel directions in spin space in the ground state. The ground-state energy is minimal and the spin configuration is collinear already at the Ising limit. Perhaps the simplest structures associated with frustration are polygons with an odd number of vertices with edges that correspond to exchange interactions of the same strength. The smallest member of this family is the triangle, for which in the classical Ising ground state not all three spin pairs can be simultaneously antiparallel, demonstrating the effect of frustration. Going to the XY model frustration is relaxed by allowing the spins to interact in a plane, and the lowest-energy configuration becomes coplanar and its energy is continuously lowered with increasing coupling in the second spin direction. This holds up to the XX limit, where the ground-state energy assumes its minimum value and does not decrease any further if interaction in the third spin direction is introduced. The energy per bond is then equal for any of the three bonds. Analogous results hold for any member of the family of the polygons with an odd number of vertices, even though frustration gets weaker with size [9]. The minimum dimensionality of the ground states (MDGS[1] ) in spin space is equal to the real-space dimensionality of the polygon.

A basic question for molecules including at least one type of odd-membered polygon is if the assembly of more than one frustrated units further strengthens frustration, by increasing the ground-state energy per bond and making the MDGS noncoplanar. For the icosidodecahedron, an Archimedean solid with vertex-sharing triangles [10–14], this is not true, as the

---

[1]The absolute ground-state energy may correspond to configurations in different number of spin dimensions. Here the configuration with minimum dimensionality in spin space (MDGS) is found.

ground-state energy is simply the sum of the energies of the individual triangles and the corresponding MDGS is coplanar [15]. The same is true for the ground-state energies of two other Archimedean solids [16], the truncated tetrahedron and the truncated icosahedron [17–22], however the corresponding MDGSs are noncoplanar [23,24]. Their frustrated units are triangles and pentagons respectively, which are isolated from one another. On the other hand for the icosahedron, a Platonic solid with edge-sharing triangles [25–27], not only is the MDGS three-dimensional, but also the corresponding energy per bond is higher than the one of an isolated triangle [9,28,29]. A similar result holds for another Platonic solid, the dodecahedron, which is formed from edge-sharing pentagons [30].

## 1.2   Classical $n$-vector Model

In this paper magnetic frustration is characterized by calculating the minimum spin-space dimensionality of the absolute ground state of the nearest-neighbor classical $n$-vector model, when it describes interactions between spins with $n$ components residing on vertices of frustrated molecules. For $n = 1$ one gets the Ising model, for $n = 2$ the XY model, and for $n = 3$ the Heisenberg model, but $n$ is allowed to be greater than three until the ground-state energy does not decrease any more. In this way the spins are given more space for nearest neighbors to attempt to direct themselves in an antiparallel fashion, not being restricted by the three dimensions of the Heisenberg model but only by the frustrated topology of the underlying structure. The ground-state energy is monitored from the Ising limit up to the minimum dimension where it assumes its absolute minimum. This is done in a continuous way, allowing the detection of dimensionality windows where the lowest energy remains constant before it starts to decrease again. The point-group symmetry of the molecule determines the number of independent exchange interactions, with edges connected by symmetry operations taken to correspond to the same interaction strength. A second way with which the strength of magnetic frustration is characterized is by calculating the maximum possible ground-state energy, when there is more than one independent exchange interaction and these are allowed to vary in the minimum number of spin dimensions of the absolute ground state. In this paper molecules having up to four unique exchange interactions are considered.

The task to investigate the properties of the $n$-vector model when $n > 3$ has been mostly undertaken in the continuum case [31–41]. Here Platonic solids in three and four spatial dimensions are considered, as well as Archimedean solids, fullerene molecules [42,43], and geodesic icosahedra [44,45]. These molecules include at least one type of odd-membered polygons.

It is found that for the icosahedral Platonic solids and their fourth-dimensional real space analogues (Table 1) the MDGS is equal to the real-space dimensionality of the molecule, showing a direct correlation between spin space and real space. Frustration is stronger than the one of the isolated polygon these molecules are made from, since the ground-state energy per bond for them is higher. On the other hand the truncated dodecahedron and the rhombicosidodecahedron, which are Archimedean solids (Table 2), have three-dimensional MDGSs and frustration is not stronger than the one at the level of the isolated polygons they are made of. However in the case of the snub dodecahedron the assemblage of the individual polygons increases the maximum possible frustration.

For the fullerene molecules (Table 2) the MDGS is typically three or four-dimensional, with the ones with icosahedral symmetry belonging to the former case. Frustration is minimal for them, as the maximum ground-state energy does not exceed the ground-state energy of an isolated pentagon. This demonstrates a direct correlation between magnetic behavior and symmetry and shows that frustration does not necessarily decrease with the size of the fullerenes, but rather there is a specific symmetry that minimizes it. Molecules with non-isolated pentagons and $T_d$ symmetry have MDGSs in $n = 5$. $T_d$ fullerenes exhibit common

magnetic behavior not only in the minimum dimensionality of the absolute ground state, but also in the correlations of the maximally frustrated ground state. Fullerenes more generally show that the maximum ground-state energy per bond does not necessarily decrease with the number of vertices, showing that symmetry is also important in determining the strength of frustration, as was also pointed out earlier for the icosahedral members of the family.

The minimum dimensionality in which the smallest geodesic icosahedron, the pentakis dodecahedron, develops its absolute ground state is $n = 4$ (Table 2). Its maximum ground-state energy per bond equals the ground-state energy per bond of the icosahedron, even though it develops in four dimensions. The ground state of the next bigger member of the family, the pentakis icosidodecahedron, develops in $n = 5$, and its maximum possible energy per bond value is higher than the one of the icosahedron. The pentakis snub dodecahedron, which lacks a center of inversion, has a three-dimensional MDGS. Its maximum energy value is achieved when two of the exchange interactions are zero and is also equal to the ground-state energy per bond of the icosahedron. The next bigger member of the family, the hexapentakis truncated icosahedron, has a five-dimensional MDGS. Its maximum energy value is again achieved when two of the exchange interactions are zero and is again equal to the ground-state energy per bond of the icosahedron, also existing in three dimensions. The maximum ground-state energy per bond of the geodesic icosahedra also shows that frustration does not necessarily decrease with the number of vertices, with the pentakis icosidodecahedron having the highest one. Since these molecules, except for the pentakis snub dodecahedron, have $I_h$ spatial symmetry, what makes the difference is their connectivity.

A common characteristic of the lowest-energy configuration in a spin-space dimension $n$ less than the one of the MDGS $n_g$, is that it typically does not change before the interactions that start to develop in the next higher dimension assume a minimum strength. Unlike odd-membered polygons which are simple frustrated structures, the ground-state energy does not continuously decrease as the interaction in the new dimension is getting stronger. The minimum interaction strength required to lower the ground-state energy typically increases with dimensionality, showing that the lowest-energy configuration is becoming more stable with $n$. Similarly, less strongly frustrated ground states for a specific $n_g$ require a stronger minimum interaction strength in the new dimension in order to lower their energy.

The plan of this paper is as follows: in Sec. 2 the $n$-vector model is introduced, and in Secs 3 to 8 its MDGS for different families of molecules is calculated. Finally Sec. 9 presents the conclusions.

## 2   Model

$N$ classical spins $\vec{s}_i$, $i = 1, \ldots, N$ of unit magnitude are considered, defined in an $n$-dimensional spin space. A single spin is mounted on each of the $N$ vertices of the different molecules under consideration. The spins interact according to the Hamiltonian $H_n$ of the nearest-neighbor $n$-vector model, with two interacting spins connected by an edge of the molecule. The exchange interactions are taken to obey the molecular symmetry, with two edges connected by a symmetry operation of the molecular point group corresponding to the same interaction strength. An interaction between nearest neighbors $i$ and $j$ is taken equal to $J_{ij}$ in $n - 1$ of the spin directions, and is scaled with $\alpha_n$ in the remaining one:

$$H_n = \sum_{<ij>} J_{ij} \left( \sum_{\sigma=1}^{n-1} s_i^\sigma s_j^\sigma + \alpha_n s_i^n s_j^n \right). \tag{1}$$

The brackets indicate that interactions are limited to nearest neighbors. $J_{ij}$ is nonnegative and $0 < \alpha_n \leq 1$, as the dimensionality of spin interactions $d = n - 1 + \alpha_n$ goes from $n - 1$

to $n$ in spin space. For bipartite structures nearest neighbors point in antiparallel directions in the ground state, while for frustrated ones the topology of the molecule determines their relative orientation. Here the lowest-energy configuration of Hamiltonian (1) as $n$ is varied is of interest. The lowest possible dimensionality is the Ising limit $n = 1$, and $n$ is allowed to increase until the ground state does not change any further. When $n = 2$ and $0 < \alpha_2 < 1$ the Hamiltonian is the one of the XY model, and at $n = 2$ and $\alpha_2 = 1$ of the XX model. For $n = 3$ and $0 < \alpha_3 < 1$ one gets the XXZ model, and for $n = 3$ and $\alpha_3 = 1$ the XXX model.

The number of independent exchange interactions grows with molecular size and decreasing symmetry. A thorough investigation of interaction space has been made for molecules with up to four unique interactions. These are parametrized so that the sum of the squares of their magnitudes equals one. If their number is two they are parametrized by a polar angle $\omega$, and their values are equal to the corresponding $x$ and $y$ components. If it is three a polar $\theta$ and an azimuthal angle $\phi$ do the parametrization, with the three different interactions equal to the corresponding $x$, $y$, and $z$ components.

The calculations were done numerically [24, 46, 47]. Each spin $\vec{s}_i$ being a classical unit vector in $n$ dimensions is defined by $n - 2$ polar angles varying from 0 to $\pi$ and an azimuthal angle varying from 0 to $2\pi$. A random initial configuration of the spins is selected and each angle is moved opposite its gradient direction, until the lowest energy configuration is reached. Repetition of this procedure for different initial configurations generates the energy minimum within the numerical accuracy of the calculation [48–50].

# 3 Polygons with an Odd Number of Vertices

The lowest-energy configuration of model (1) on bipartite structures, which lack any frustration, is of the Néel type, with nearest-neighbor spins pointing in antiparallel directions. It is already the ground state at the Ising limit of Hamiltonian (1).

A simple family of frustrated structures are polygons with an odd number of vertices. In the simplest case all polygon edges are equivalent and $J_{ij} \equiv J$ in Hamiltonian (1), with each spin having two nearest neighbors. In the Ising-limit ground state neighboring spins are antiparallel, except from a single parallel pair due to the odd total number of spins. Frustration is getting weaker with $N$ as the number of antiparallel nearest-neighbor pairs increases. As interaction in the second direction is introduced by allowing $\alpha_2$ to be nonzero, spin-space anisotropy and consequently frustration are getting weaker. This results in a two-dimensional MDGS in contrast to the unfrustrated case, and a continuously decreasing energy per bond and net magnetization with increasing $d$ (Fig. 1). At the XX limit the angle between nearest-neighbors equals $\frac{N-1}{N}\pi$ [9], while the net magnetization becomes zero. Further increasing $d$ in the Hamiltonian does not decrease the ground-state energy, and the dimensionality of the MDGS in spin space equals the real-space dimensionality of the polygons.

# 4 Platonic Solids

## 4.1 Icosahedron

The icosahedron belongs to the class of Platonic solids [25], which are convex regular polyhedra with equivalent vertices that consist of only one type of polygon. Their edges are also equivalent and each $J_{ij} \equiv J$ in Hamiltonian (1). The icosahedron consists of 20 triangles, has 12 vertices, and belongs to the $I_h$ symmetry group, the point group with the largest number of symmetry operations [51]. At the XXX limit the ground state has been found to be noncopla-

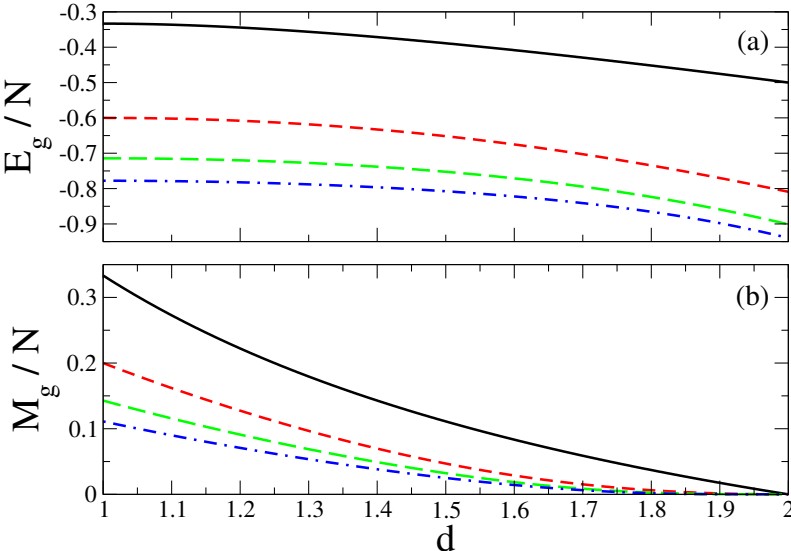

Figure 1: (a) Ground-state energy per bond $\frac{E_g}{N}$ for Hamiltonian (1) for a triangle ((black) solid line), a pentagon ((red) dashed line), a heptagon ((green) long-dashed line), and a nonagon ((blue) dot-dashed line) as a function of the dimensionality $d$ of the interactions in spin space. (b) Corresponding magnetization per spin $\frac{M_g}{N}$.

nar [9, 28, 29]. The triangles share edges resulting in stronger frustration in comparison with an isolated triangle, as the ground-state nearest-neighbor correlation increases from $-\frac{1}{2}$ in the latter case to $-\frac{\sqrt{5}}{5} = -0.44721$. Fig. 2(a) shows the evolution of the ground-state energy per bond as $d$ varies from 1 to 3. At the Ising limit it equals $-\frac{1}{3}$, as in the case of an isolated triangle. Unlike the triangle and the other odd-membered polygons though (Fig. 1(a)), the lowest-energy Ising configuration does not immediately change when $\alpha_2$ becomes nonzero, remaining the same up to $\alpha_2 = \frac{\sqrt{5}}{5}$. This can also be seen in the plot of the nearest-neighbor correlations as a function of $d$ (Fig. 3(a)). The lowest-energy configuration becomes then noncollinear with several unique nearest-neighbor correlation values, until the XX limit where nearest-neighbors are antiparallel or at angles $\frac{\pi}{3}$ and $\frac{2\pi}{3}$ with each other, and the ground-state energy per bond equals $-\frac{2}{5}$. The XX limit ground state is also protected against $\alpha_3$ as it does not change up to $\alpha_3 = 0.47178$, where it becomes noncoplanar. Finally at the XXX limit the nearest-neighbor correlations become the same for every pair. The ground-state energy does not change as $d$ is further increased, and $d = 3$ is the minimum spin-space dimensionality for which all nearest-neighbor correlations become equal and obey the symmetry of the icosahedron. The total spin equals zero for any value of $d$.

## 4.2 Dodecahedron

The dodecahedron, also a Platonic solid with $I_h$ symmetry, consists of 20 edge-sharing pentagons and has 20 vertices [52–54]. Like the icosahedron, the ground state of model (1) at the XXX limit is three-dimensional, with each bond having an energy equal to $-\frac{\sqrt{5}}{3} = -0.74536$ [30]. This is higher than the energy per bond in the coplanar ground state of the isolated pentagon $-\frac{\sqrt{5}+1}{4} = -0.80902$ (Fig. 1(a)) [9]. Fig. 2(b) shows how the ground-state energy per bond changes as $d$ goes from from 1 to 3. The Ising energy per bond is $-\frac{3}{5}$, equal to the one of an isolated pentagon, but unlike the latter (Fig. 1(a)) and similarly to the icosahedron, it takes a finite $\alpha_2 = 0.48733$ for the Ising ground state to give way to a lower-energy configuration

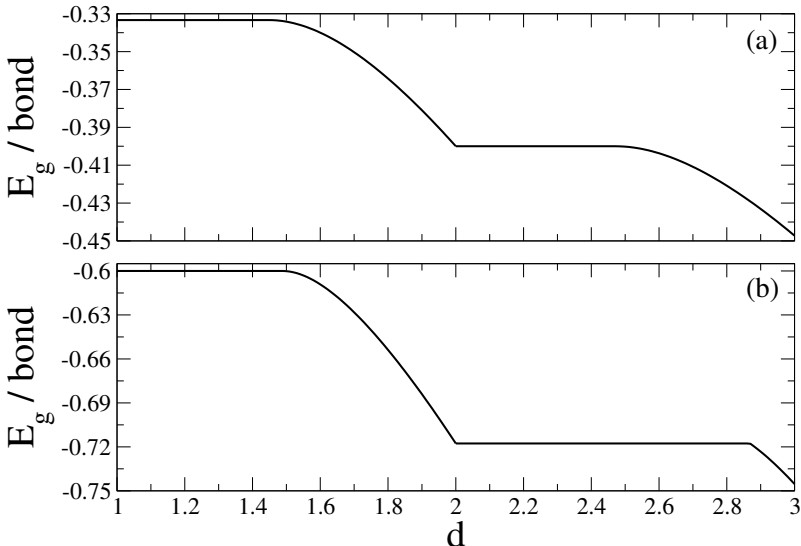

Figure 2: Ground-state energy $E_g$ per bond for Hamiltonian (1) as a function of the dimensionality $d$ of the interactions in spin space for (a) the icosahedron and (b) the dodecahedron.

which is noncollinear and has several unique nearest-neighbor correlations values (Fig. 3(b)). A value of $\alpha_3 = 0.86857$ is required for the ground state to change to a noncoplanar configuration with a discontinuous derivative of the ground-state energy with respect to $d$. This eventually becomes the configuration with all nearest-neighbor correlations equal at the XXX limit and the energy does not change if $d$ is further increased. Similarly to the icosahedron, the equality of all the lowest-energy configuration's nearest-neighbor correlations agrees with the geometrical equivalence of all of the dodecahedron's edges. Other similarities are that the minimum $\alpha_3$ value required to lower the ground-state energy is bigger than the corresponding $\alpha_2$, and that the total spin is zero for any $d$. The icosahedron and dodecahedron's MDGS of Hamiltonian (1) is found in a number of spin dimensions equal to their real-space dimensionality (Table 1).

## 5 Platonic Solids in Four Spatial Dimensions

The icosahedron and the dodecahedron analogues in four spatial dimensions are the 600-cell and the 120-cell respectively [55]. The 600-cell has 120 vertices, 720 edges, and 1200 triangular faces. Fig. 4 shows the evolution of the ground-state energy of Hamiltonian (1) as a function of $d$, as well as the value of the ground-state magnetization per spin when it is nonzero. Introducing the interaction in the second spin direction away from the Ising limit results in an immediate lowering of the energy unlike the three-dimensional Platonic solids, and $a_2 = 0_+$. The $d = 2$ and 3 ground states again do not change with the introduction of interactions in the third and fourth dimension respectively before $\alpha_n$ gets sufficiently strong, and $a_3 = 0.68505$. When $d = 3$ the average energy per bond equals $-0.29098$, and at $d = 4$ it is equal to $-\frac{\sqrt{5}-1}{4} = -0.30902$ for every bond, significantly higher than the value of the icosahedron. This shows that frustration is getting quite stronger when going from three to four-dimensional spatial dimensions, and that the dimensionality of the MDGS follows the spatial dimensionality of the molecule (Table 1), with nearest-neighbor correlations becoming equal exactly at $d = 4$. The value of $a_4$ is 0.62348, less than $a_3$, again unlike the three-

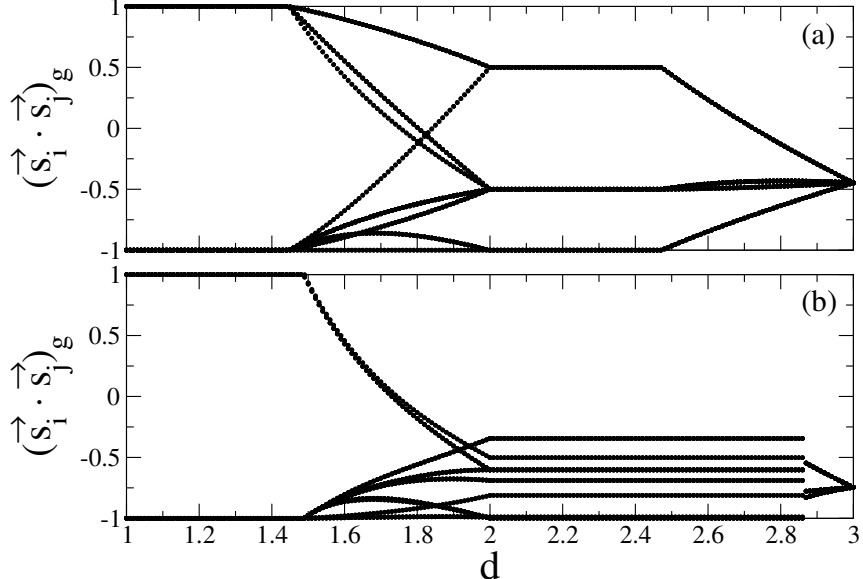

Figure 3: Ground-state nearest-neighbor correlations $(\vec{s}_i \cdot \vec{s}_j)_g$ for Hamiltonian (1) as a function of the dimensionality $d$ of the interactions in spin space for (a) the icosahedron and (b) the dodecahedron.

dimensional Platonic solids. The ground-state magnetization per spin has discontinuities in its value and also in its derivative as a function of $d$.

The four-dimensional analogue of the dodecahedron, the 120-cell, has 600 vertices, 1200 edges, and 720 pentagonal faces. Its ground-state energy per bond evolution from $d = 3$ to 4 is shown in Fig. 5. The average ground-state nearest-neighbor correlation when $d = 3$ equals $-0.69752$. The absolute minimum of the ground-state energy is again firstly achieved when $d = 4$, equal to $-\frac{3\sqrt{5}-1}{8} = -0.71353$ for every bond (Table 1), higher than the corresponding value for the dodecahedron. Again only at the absolute minimum of the ground-state energy do all nearest-neighbor correlations become equal. The value of $a_4$ is 0.88920. At this value the magnetization per spin drops from $3.2314 \times 10^{-4}$ to zero.

## 6   Archimedean Solids

The Archimedean solids are convex uniform polyhedra that have identical vertices like the Platonic solids, but are formed by more than one polygon. The icosidodecahedron is made of triangles and pentagons, with like polygons sharing vertices and unlike polygons sharing edges, and has $I_h$ symmetry and identical edges. The ground-state energy per bond of Hamiltonian (1) equals the one of an isolated triangle, with the lowest-energy configuration similarly being coplanar, showing that it is determined by the more strongly frustrated triangles [15].

The truncated tetrahedron and the truncated icosahedron, which have $T_d$ and $I_h$ symmetry respectively, have been shown to have ground states three-dimensional in spin space for any ratio of their two symmetrically independent exchange constants [23, 24]. They achieve the lowest-possible energy allowed by the connectivity of the frustrated polygon they include, the triangle and the pentagon respectively, which is the sum of energies of isolated frustrated polygons and antiparallel interpolygon bonds.

The MDGSs of the truncated dodecahedron and the rhombicosidodecahedron, which also have $I_h$ symmetry and two symmetrically independent exchange constants, are also

Table 1: Different families of molecules with one symmetrically independent exchange interaction for which the ground state of Hamiltonian (1) was calculated. The real-space dimensionality of the molecules is listed, along with the number of vertices $N$, the point-group symmetry, the minimum ground-state dimensionality in spin space $n_g$, and the ground-state energy per bond $\frac{E_g}{bond}$.

| molecule | real-space dimensions | $N$ | point-group symmetry | $n_g$ | $\frac{E_g}{bond}$ |
|---|---|---|---|---|---|
| Polygons with an Odd Number of Vertices | | | | | |
| triangle | 2 | 3 | $D_3$ | 2 | $-\frac{1}{2}$ |
| pentagon | 2 | 5 | $D_5$ | 2 | $-\frac{\sqrt{5}+1}{4}$ |
| Platonic Solids | | | | | |
| icosahedron | 3 | 12 | $I_h$ | 3 | $-\frac{\sqrt{5}}{5}$ |
| dodecahedron | 3 | 20 | $I_h$ | 3 | $-\frac{\sqrt{5}}{3}$ |
| Four-Dimensional Platonic Solids | | | | | |
| 600-cell | 4 | 120 | $H_4$, [3,3,5] | 4 | $-\frac{\sqrt{5}-1}{4}$ |
| 120-cell | 4 | 600 | $H_4$, [3,3,5] | 4 | $-\frac{3\sqrt{5}-1}{8}$ |

three-dimensional and achieve the lowest-possible ground-state energy allowed by their odd-membered polygons for any relative value of the exchange constants. In the former case the energy is the sum of energies of isolated triangles and antiparallel intertriangle bonds, while in the latter of isolated triangles and pentagons. Consequently they achieve the maximum possible energy when only the triangle bonds are nonzero, with the corresponding MDGS two-dimensional in spin space (Table 2).

The snub dodecahedron has chiral icosahedral symmetry $I$, lacking a center of inversion. Unlike the aforementioned Archimedean solids it has edge-sharing frustrated units of the same type, which are triangles. It has three symmetrically independent edges, which divide it into pentagons, triangles, and dimers, since not all triangle bonds are symmetrically equivalent. Figure 6 shows the reduced ground-state energy per bond $\frac{E_g}{\sum_{<ij>} J_{ij}}$ when the relative strengths of the corresponding three unique exchange interactions are parametrized by a polar angle $\theta$ and an azimuthal angle $\phi$. The $z$-component corresponds to the exchange interaction of the pentagon edges, the $x$ to one of the triangles, and the $y$ to the one of the dimers. The MDGS is again three-dimensional in spin space, with the nearest-neighbor correlations assuming three independent values, corresponding to the unique exchange interactions. The total spin of the ground state is zero. For $\theta = 0$ only the pentagon bonds are nonzero and the ground-state energy is the sum of energies of isolated pentagons. For $\theta = \frac{\pi}{2}$ only the triangle and dimer bonds are nonzero. As $\phi$ increases from 0 to $\frac{\pi}{2}$ the energy goes from the sum of energies of isolated triangles to the sum of energies of isolated dimers. For $0 < \theta < \frac{\pi}{2}$ the ground-state energy is determined by the competition of the three different exchange constants. The competition between the triangular and pentagonal bonds increases the energy for smaller $\phi$ when $\theta \neq \frac{\pi}{2}$, while for higher $\phi$ the dimer bonds, which link different pentagons, are getting stronger and lower the ground-state energy.

Table 2: Different families of three-dimensional molecules with more than one symmetrically independent exchange interaction for which the MDGS of Hamiltonian (1) was calculated for the whole range of their exchange interactions parameter space. The number of vertices of the molecules $N$ is listed, along with the point-group symmetry, the number of symmetrically independent exchange interactions, the minimum ground-state dimensionality in spin space $n_g$, and the maximum possible reduced ground-state energy per bond $(\frac{E_g}{\sum_{<ij>} J_{ij}})_{max}$ along with the minimum corresponding dimensionality in spin space $n_{max}$.

| molecule | $N$ | point group | unique inter. | $n_g$ | $(\frac{E_g}{\sum_{<ij>} J_{ij}})_{max}$ | $n_{max}$ |
|---|---|---|---|---|---|---|
| Archimedean Solids | | | | | | |
| truncated dodecahedron | 60 | $I_h$ | 2 | 3 | $-\frac{1}{2}$ | 2 |
| rhombicosidodecahedron | 60 | $I_h$ | 2 | 3 | $-\frac{1}{2}$ | 2 |
| snub dodecahedron | 60 | $I$ | 3 | 3 | -0.4863181 | 3 |
| Fullerenes | | | | | | |
| chamfered dodecahedron | 80 | $I_h$ | 2 | 3 | $-\frac{\sqrt{5}+1}{4}$ | 2 |
| hexpropello dodecahedron | 140 | $I$ | 4 | 3 | $-\frac{\sqrt{5}+1}{4}$ | 2 |
| truncated pentakis dodecahedron | 180 | $I_h$ | 4 | 3 | $-\frac{\sqrt{5}+1}{4}$ | 2 |
| - | 24 | $D_{6d}$ | 3 | 3 | -0.7606899 | 2 |
| - | 28 | $T_d$ | 3 | 4 | -0.7843647 | 4 |
| - | 30 | $D_{5h}$ | 4 | 3 | -0.7707600 | 3 |
| - | 36 | $D_{6h}$ | 4 | 2 | $-\frac{\sqrt{5}+1}{4}$ | 2 |
| - | 40 | $T_d$ | 4 | 5 | -0.7843647 | 3 |
| Geodesic Icosahedra | | | | | | |
| pentakis dodecahedron | 32 | $I_h$ | 2 | 4 | $-\frac{\sqrt{5}}{5}$ | 4 |
| pentakis icosidodecahedron | 42 | $I_h$ | 2 | 5 | -0.4403875 | 5 |
| pentakis snub dodecahedron | 72 | $I$ | 4 | 3 | $-\frac{\sqrt{5}}{5}$ | 3 |
| hexapentakis truncated icosahedron | 92 | $I_h$ | 4 | 5 | $-\frac{\sqrt{5}}{5}$ | 3 |

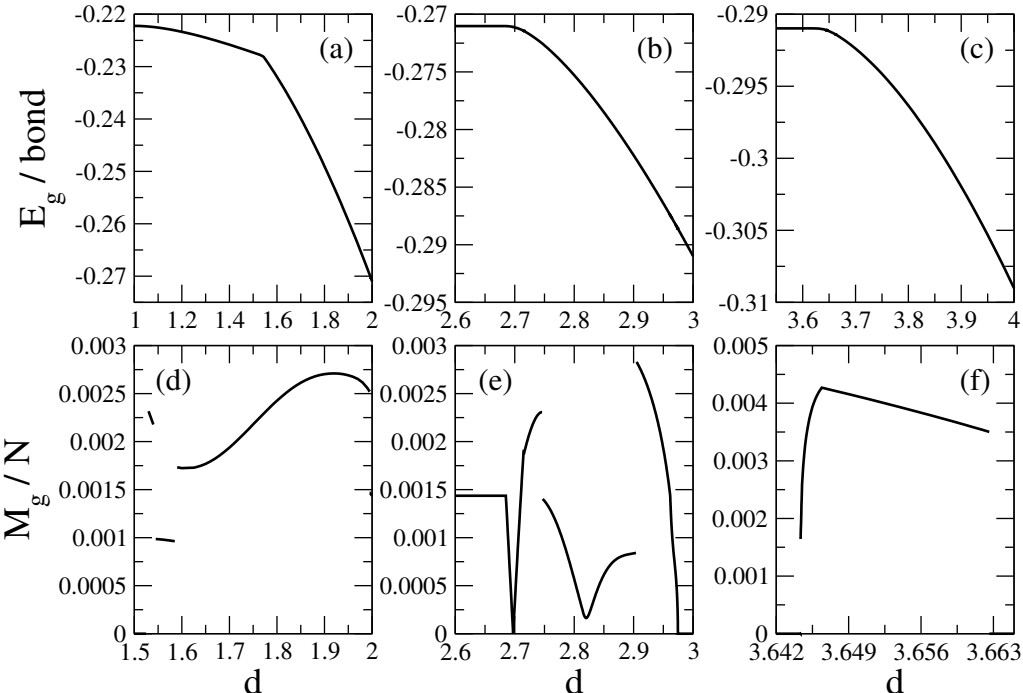

Figure 4: (a), (b), and (c): ground-state energy $E_g$ per bond and (d), (e), and (f): ground-state magnetization per spin $\frac{M}{N}$ for Hamiltonian (1) as a function of the dimensionality $d$ of the interactions in spin space for the 600-cell.

Frustration has a peak when the lowest energy becomes maximum. This occurs when $\theta = 0.379280\pi$ and $\phi = \frac{\pi}{4}$, where the triangle and dimer interactions are both equal to 0.65686 while the pentagonal is equal to 0.37023. The nearest-neighbor correlations are equal to -0.4863181 for every bond (Table 2), higher than the corresponding value for an isolated triangle but lower than the value for the icosahedron. For $\theta$ not very close to $\frac{\pi}{2}$ and $\phi \approx 0.481\pi$, where the dimer bonds are strong and the triangle bonds weak, minimization of the pentagonal energy is favored and the reduced energy per bond is equal to the one of an isolated pentagon. The equality occurs for slightly different values of $\phi$ as $\theta$ is varied.

Archimedean solids have MDGSs which are at most three-dimensional (Table 2), and frustration is not stronger than the one at the level of an isolated odd-membered polygon. The exception is the snub dodecahedron, which achieves a maximally frustrated ground state with an energy per bond higher than the one of an isolated triangle. Simultaneously, and even though there are three geometrically distinct edges in the molecule, the maximally frustrated ground state achieves the same energy for any bond in the molecule.

# 7 Fullerene Molecules

Fullerene molecules are allotropes of carbon that consist of 12 pentagons and $\frac{N}{2}-10$ hexagons [42, 43]. The polygons share edges and their vertices are three-fold coordinated. Members of the family are the dodecahedron (Sec. 4.2) and the truncated icosahedron (Sec. 6). The pentagons are the source of frustration, and the molecules are further characterized by having neighboring (nonisolated) pentagons or not. A three-dimensional ground state has been found to be a typical feature of fullerene molecules when spins mounted on their vertices interact according to the XXX model and all exchange interactions are equal [24, 46, 56–58]. Frustra-

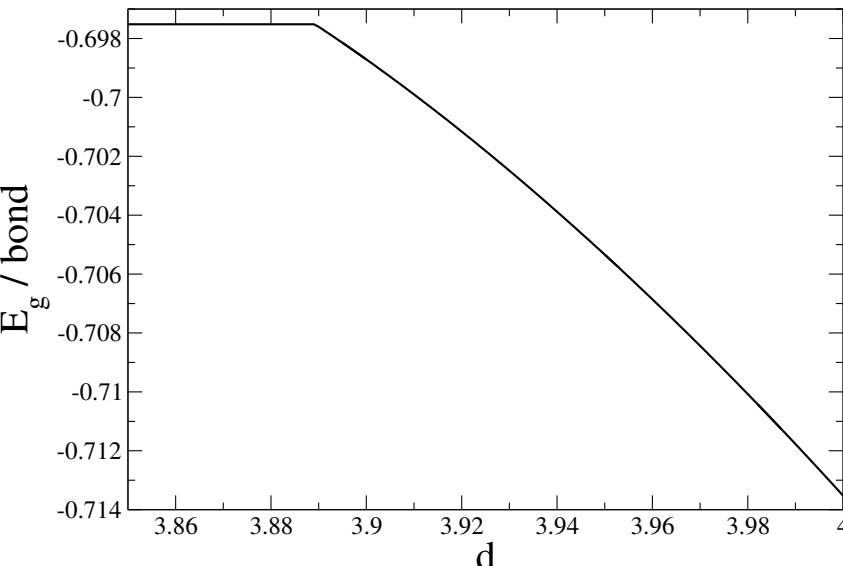

Figure 5: Ground-state energy $E_g$ per bond for Hamiltonian (1) as a function of the dimensionality $d$ of the interactions in spin space for the 120-cell.

tion even results in a finite ground-state magnetization for many of the molecules. On average frustration decreases with the number of vertices as the number of hexagons increases, but a more precise characterization of frustration, especially for molecules with the same number of vertices, can be done if $n$ in Hamiltonian (1) is allowed to be greater than 3 and the exchange interactions can take arbitrary values. The interaction parameter space has been investigated for fullerene molecules that have up to four symmetrically independent exchange interactions.

$I_h$-symmetry molecules bigger than the dodecahedron have pentagons not neighboring one another. For the molecules with 80 (chamfered dodecahedron) and 180 (truncated pentakis dodecahedron) vertices and the $I$-symmetry molecule with 140 (hexpropello dodecahedron) the MDGS is three-dimensional. The corresponding average energy per bond varies between the one of an isolated pentagon and that of antiparallel spins, meaning that the maximum possible energy occurs when only the pentagons bonds are nonzero and the corresponding MDGS is two-dimensional (Table 2). This shows that for icosahedral symmetry frustration is never stronger than the one of an isolated pentagon, as has already been found for the truncated icosahedron [24], and demonstrates a strong correlation between symmetry and magnetic behavior. It has also been found that the icosahedral fullerene molecules have the same classical magnetization response and are also the ones that support magnetization discontinuities for quantum spins [46, 56, 59].

More specifically, for $N = 80$ there are two symmetrically unique types of edges, the first between same-pentagon spins and the second between a pentagon and a nonpentagon spin. The reduced ground-state energy per bond is shown in Fig. 7 as a function of $\omega$, with $\tan \omega$ the relative strength of the two exchange interactions and for $d$ ranging from 1 to 3. Also shown is the lower bound for the energy, which is the sum of the ground-state energies of the corresponding number of isolated pentagons and hexagons. Unlike the truncated icosahedron [24], the chamfered dodecahedron does not attain this lower bound, which explains why the intrapentagon and pentagon-nonpentagon nearest-neighbor correlations vary with $\omega$ and are not constant and equal to $-\frac{\sqrt{5}+1}{4}$ and -1 respectively. However the maximum frustration does not exceed the one of an isolated pentagon, with the reduced ground-state energy per bond never higher than $-\frac{\sqrt{5}+1}{4}$. This is also true for the icosahedral molecules with $N = 140$ and 180,

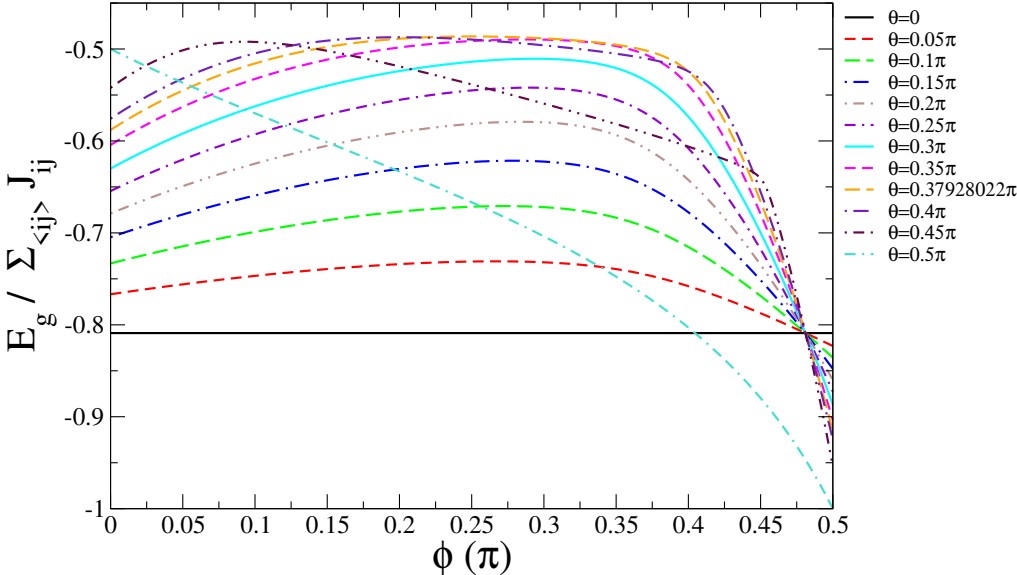

Figure 6: Reduced ground-state energy per bond $\frac{E_g}{\sum_{<ij>} J_{ij}}$ for Hamiltonian (1) in $d = 3$ for different values of $\theta$ as a function of $\phi$ for the snub dodecahedron. $\theta$ and $\phi$ control the relative strength of the three symmetrically independent exchange constants, with the $z$-component the exchange interaction of the pentagon edges, the $x$ the one of the triangles, and the $y$ the one of the dimers.

which have four symmetrically unique types of edges, showing that the icosahedral clusters achieve minimal frustration. Frustration does not have to do with the size of the molecule and the fact that the number of unfrustrated hexagons increases with $N$, but rather with molecular symmetry.

For 24 vertices and $D_{6d}$ symmetry [42, 60] the MDGS of Hamiltonian (1) is in general three-dimensional (Fig. 8) and has zero magnetization. The maximum energy per bond equals -0.7606899 when the exchange interactions in the top and bottom hexagon equal $J_1 = 0.369747$, the ones between spins in the top or bottom hexagon and spins in the middle ring equal $J_2 = 2J_1$, and the ones between spins in the middle ring equal $J_3 = 0.562526$, with the ground state developing then minimally in two spin dimensions. The $J_1$ and $J_2$ bonds have an energy equal to $-\frac{J_3}{2J_1}$, which equals the ground-state energy per bond, while in the middle ring the energy alternates in value between -1 and $1 - \frac{J_3}{J_1}$, so that the average also equals $-\frac{J_3}{2J_1}$.

For 28 vertices and $T_d$ symmetry [42,60] the MDGS is four-dimensional (Fig. 9). Maximum frustration occurs when same-hexagon exchange interactions are equal to $J_1 = 0.362873$, interhexagon interactions to $J_2 = 2J_1$, and the rest of the pentagon-only interactions to $J_3 = 0.584480$. All nearest-neighbor correlations are equal to -0.7843647, and the magnetization per spin equals 0.0599442. The exchange interaction values for the maximum energy per bond are not very different from the ones of the 24-vertices cluster, and similarities can also be detected between the $\theta$ and $\phi$ dependence in Figs 8 and 9, with the number of hexagons much less than the one of pentagons for the two clusters.

For 30 vertices and $D_{5h}$ symmetry [42, 60] the MDGS is three-dimensional. Its energy is maximized when the exchange interactions that connect the lower with the upper part of the molecule are zero, dividing it in two parts with respect to its mirror plane. The interaction within the top and bottom pentagons equals 0.73198, between these pentagons and the hexagons 0.59805, and hexagon spins within the upper or the lower side of the molecule

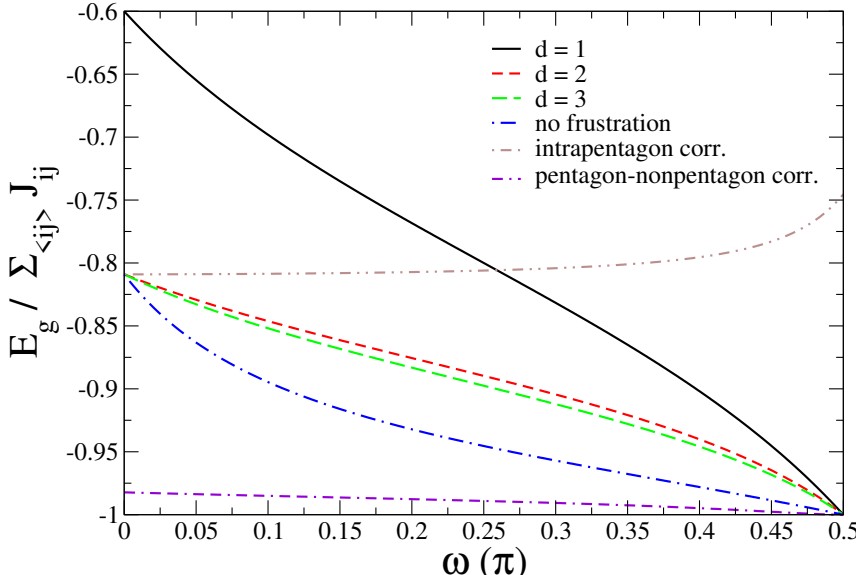

Figure 7: Reduced ground-state energy per bond $\frac{E_g}{\sum_{<ij>} J_{ij}}$ for Hamiltonian (1) as a function of $\omega$ for the chamfered dodecahedron ($N = 80$), with $cos\omega$ the intrapentagon and $sin\omega$ the pentagon-nonpentagon interaction. The dimensionality of spin space $d$ ranges from 1 to 3. "No frustration" shows the sum of reduced ground-state energies of isolated pentagons and hexagons. The intrapentagon and pentagon-nonpentagon nearest-neighbor correlation functions are also plotted.

interact with bond strengths equal to 0.32641. Again these values are similar to the two fullerene molecules examined before. All nearest-neighbor correlations equal -0.7707600, a value higher that the one of the $T_d$ molecule with the lower number of 28 vertices. This demonstrates that even though the number of vertices increases frustration does not get weaker, according to the maximum possible ground-state energy criterion. The corresponding magnetization per spin for each half of the molecule equals 0.122080.

For 36 vertices and $D_{6h}$ symmetry [42,58] the MDGS is two-dimensional. The maximally frustrated ground state occurs again when the exchange interactions that connect the lower with the upper part of the molecule are zero, resulting in two independent parts with respect to the molecule's mirror plane. The exchange interactions in the top and bottom hexagon have half the strength of the ones that connect them to the inner hexagons, and they have equal strength with the inner-hexagon interactions. Each bond has an energy equal to the energy per bond of an isolated pentagon, demonstrating that this molecule is minimally frustrated. The magnetization per spin for each half of the molecule equals $\frac{1}{6}$.

For 40 vertices and $T_d$ symmetry [42] the MDGS is five-dimensional. The maximally frustrated ground state is minimally three-dimensional, and occurs when interactions that only belong to hexagons are zero, splitting the molecule in four parts with ten spins each. Interactions that belong to both a pentagon and a hexagon are equal to $J = 0.52746$, and pentagon-only interactions have a strength equal to $1.61070J$, similarly to the $N = 28$ molecule with the same symmetry. Furthermore, all nearest-neighbor correlations are equal to the value of the ones in the maximally frustrated configuration of that molecule, demonstrating the link between symmetry and magnetic properties. The magnetization per spin for each of the four separate parts of the molecule equals 0.0479013. Again and according to the maximum possible ground-state energy criterion, this molecule is more strongly frustrated than the $D_{6h}$ fullerene with the smaller number of 36 vertices, showing the importance of symmetry for frustration.

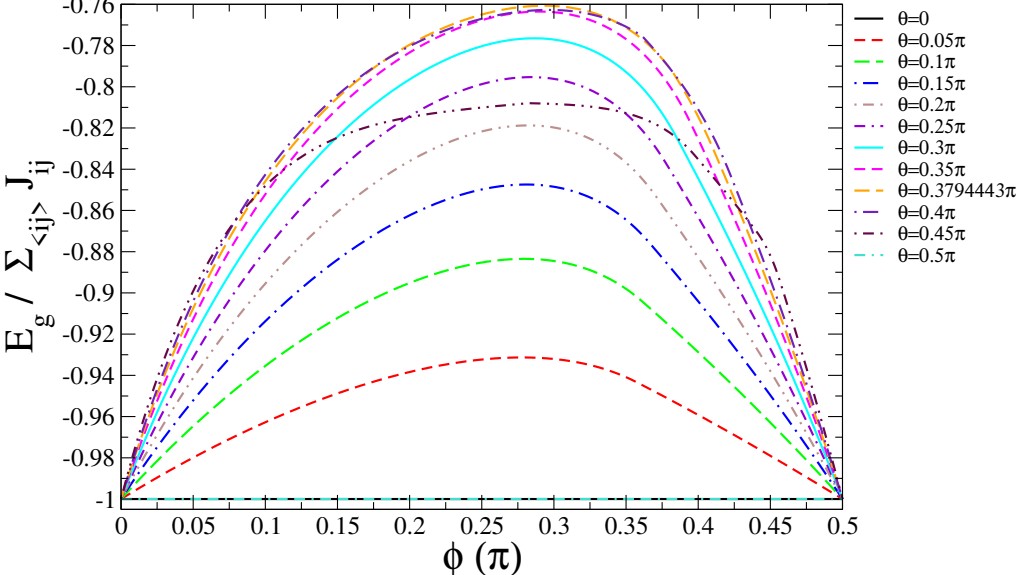

Figure 8: Reduced ground-state energy per bond $\frac{E_g}{\sum_{<ij>} J_{ij}}$ for Hamiltonian (1) in $d = 3$ for different values of $\theta$ as a function of $\phi$ for the 24-site fullerene molecule. $\theta$ and $\phi$ control the relative strength of the three symmetrically independent exchange constants, with the $z$-component the interactions in the top and bottom hexagon, the $x$ the interaction between spins in the middle ring, and the $y$ the interaction between spins in the top or bottom hexagon and spins in the middle ring.

When the number of symmetrically independent exchange interactions is greater than four, the MDGS of Hamiltonian (1) has been calculated for all $N \leq 36$ molecules and for higher symmetry molecules with $N \leq 100$ when all exchange interactions are equal, and it is typically three or four-dimensional in spin space. Along with the $N = 40$ molecule with $T_d$ symmetry, the other exception is the $N = 100$ molecule with the same symmetry and also nonisolated pentagons, which has a five-dimensional MDGS. Table 3 lists the ground-state energy of Hamiltonian (1) for both molecules with increasing $n$. The absolute minimum is achieved for $n_g = 5$, while the minimum $\alpha_n$ required to lower the ground-state energy when adding a new dimension increases with $n$, as was mostly the case for the Platonic solids.

Fullerene molecules demonstrate that even though pentagons are less frustrated than triangles, their assemblage with hexagons leads to the formation of MDGSs up to five spin dimensions (Table 2), higher even than the four of the four real-space dimensional Platonic solids. What distinguishes fullerenes from Platonic and Archimedean solids is that their vertices are not equivalent, and as $N$ increases so does the number of symmetrically unique vertices. On the other hand the icosahedral fullerenes are minimally frustrated, pointing to the importance of high symmetry for the magnetic properties, especially since they have also been found to behave nontrivially in a magnetic field both for classical and quantum spins [46, 56, 59, 61]. This is also in agreement with the icosahedral and the four-dimensional Platonic solids forming MDGSs in a number of spin dimensions equal to their dimensionality in real space, which is also true for the icosahedral Archimedean solids. These results show that frustration in fullerene molecules does not necessarily decrease with $N$ and the number of hexagons, but symmetry plays an important role as well, as is the case with the minimally frustrated $D_{6h}$ fullerene with $N = 36$. It was also found that in order to achieve maximum energy the exchange interaction values of different molecules are similar in value.

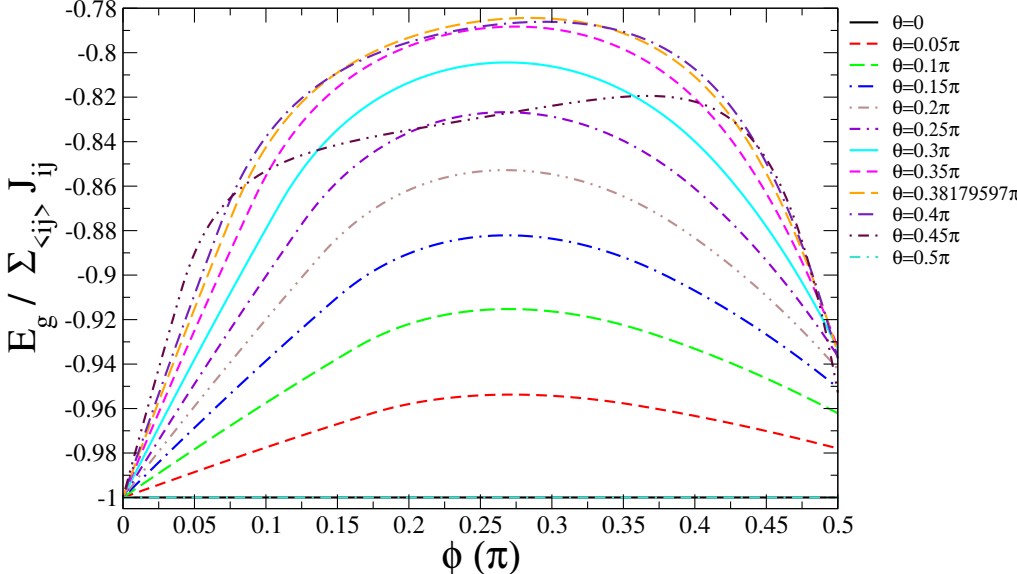

Figure 9: Reduced ground-state energy per bond $\frac{E_g}{\sum_{<ij>} J_{ij}}$ for Hamiltonian (1) in $d = 4$ for different values of $\theta$ as a function of $\phi$ for the 28-site $T_d$ fullerene molecule isomer. $\theta$ and $\phi$ control the relative strength of the three symmetrically independent exchange constants, with the $z$-component the same-hexagon interactions, the $x$ pentagon-only interactions, and the $y$ interhexagon interactions.

Table 3: Ground-state energy of Hamiltonian (1) for different $n$ at the isotropic limit $\alpha_n = 1$ and minimum $\alpha_{n,min}$ required to lower the ground-state energy in $n$ dimensions for the $T_d$-symmetry nonisolated-pentagon fullerenes with $N = 40$ and 100. All the exchange interactions are equal. The corresponding magnetizations are zero.

| | N=40 | | N=100 | |
|---|---|---|---|---|
| $n$ | $\frac{E_g}{bond}$ ($\alpha_n = 1$) | $\alpha_{n,min}$ | $\frac{E_g}{bond}$ ($\alpha_n = 1$) | $\alpha_{n,min}$ |
| 1 | $-\frac{11}{15}$ | $0_+$ | $-\frac{21}{25}$ | $0_+$ |
| 2 | -0.8265611766 | 0.48794 | -0.9163284896 | 0.47175 |
| 3 | -0.8297045764 | 0.98118 | -0.9205908313 | 0.95423 |
| 4 | -0.8298861510 | 0.99880 | -0.9206099986 | 0.99971 |
| 5 | -0.8298861626 | 0.9999998 | -0.9206100870 | 0.999995 |

## 8 Geodesic Icosahedra

The geodesic icosahedra are polyhedra derived from the icosahedron [44, 45]. They have icosahedral symmetry and are duals of fullerene molecules. Twelve of their vertices have five and the rest six nearest-neighbors and they only include triangles. The pentakis dodecahedron is the dual of the truncated icosahedron. It is derived from the dodecahedron by adding a vertex at the center of each one of its 12 pentagons, and has $N = 32$. There are two unique types of edges, one corresponding to the dodecahedron edges and the other to the edges linking a pentagon vertex with the vertex at the pentagon's center. The two corresponding exchange interactions are parametrized as $\cos\omega$ and $\sin\omega$. The zero-field ground state has been calculated for $d = 3$, along with the magnetization response both at the classical and quantum level [62]. Figure 10 shows the reduced ground-state energy per bond of Hamiltonian (1) as a function of

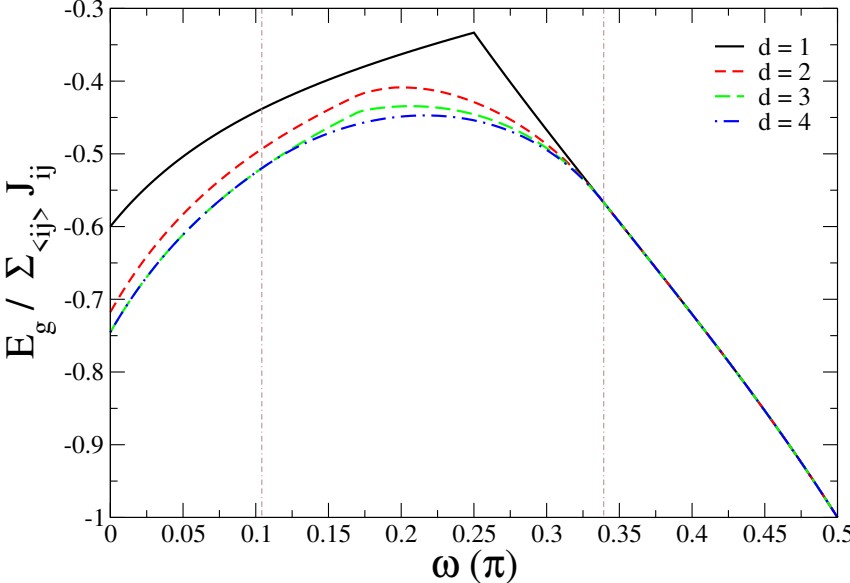

Figure 10: Reduced ground-state energy per bond $\frac{E_g}{\sum_{<ij>} J_{ij}}$ for Hamiltonian (1) as a function of $\omega$ for the pentakis dodecahedron. The dodecahedron-edge interaction equals $\cos\omega$ and the pentagon vertex with the pentagon's center interaction equals $\sin\omega$. The dimensionality of spin space $d$ ranges from 1 to 4. The vertical lines show the $\omega$ values where the dimensionality of the MDGS changes from 3 to 4 and then to 1.

$\omega$ for $d = 1$ to 4. For $\omega = 0$ the ground state of the dodecahedron is three-dimensional [30]. Introducing the second exchange interaction further enhances frustration and the reduced energy per bond increases, and eventually the MDGS develops a four-dimensional structure in spin space at $\omega = 0.10414\pi$. The maximum ground-state energy is achieved for $\omega = \tan^{-1}\frac{\sqrt{5}+1}{4}$ and equals $-\frac{\sqrt{5}}{5}$ for any bond (Table 2), as in the ground state of the icosahedron, with the corresponding magnetization per spin equal to 0.088251. This is also the maximum possible ground-state energy for spins located at the vertices of an isolated pentagon with an extra spin located at its center, whose MDGS is is three-dimensional, when the exchange strength with the spin at the center equals $\frac{\sqrt{5}+1}{2}$ times the intrapentagon interaction. At $\omega = 0.33926\pi$ the lowest-energy configuration becomes collinear.

Figure 11 shows the ground-state energy of Hamiltonian (1) as a function of $\omega$ away from the collinear regime for values of $d$ which are in general noninteger. Focusing between $d = 1$ and 2, for the ranges of $\omega$ where the ground-state energy and consequently frustration are maximum a weak $\alpha_n$ is sufficient to further lower the energy from its $d = 1$ value, while for $\omega$-ranges of weaker frustration the $d = 1$ lowest-energy configuration is more robust and the corresponding $\alpha_n$ value bigger. Going from $d = 2$ to 3 the corresponding $\alpha_n$ values are much stronger, and they get even stronger when going from $d = 3$ to 4. This shows that the lowest-energy configurations are becoming more robust with increasing $d$, a conclusion that has been typically drawn for the Platonic solids.

The ground-state nearest-neighbor correlations are shown in Fig. 12. They obey the symmetry of the molecule, with edges connected by symmetry operations corresponding to the same correlation value. They are constant for lower $\omega$ where the MDGS is three-dimensional, and for higher $\omega$ where it is collinear. In between they vary with $\omega$ and become equal at the point of maximum frustration.

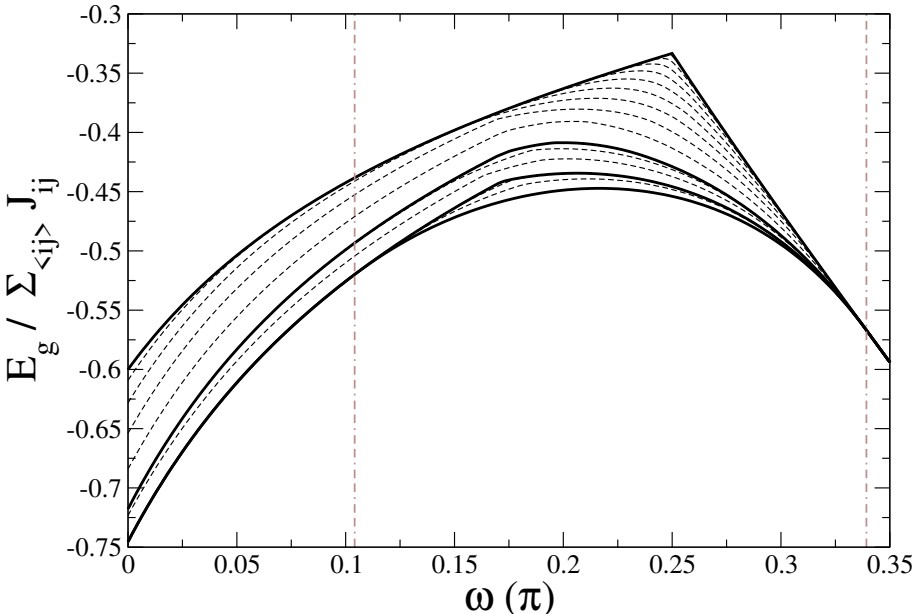

Figure 11: Reduced ground-state energy per bond $\frac{E_g}{\sum_{<ij>} J_{ij}}$ for Hamiltonian (1) as a function of $\omega$ for the pentakis dodecahedron. The dodecahedron-edge interaction equals $\cos\omega$ and the pentagon vertex with the pentagon's center interaction equals $\sin\omega$. The dimensionality of spin space $d$ ranges from 1 to 4 in steps of $\frac{1}{10}$, increasing from top to bottom, with integer-$d$ values corresponding to straight lines, and noninteger to dashed lines. Noninteger values of $d$ may coincide with integer values, especially with increasing $d$. The vertical lines show the $\omega$ values where the dimensionality of the MDGS changes from 3 to 4 and then to 1.

The next-bigger geodesic icosahedron is the pentakis icosidodecahedron. It is derived from the icosidodecahedron by introducing a vertex at the center of each one of its 12 pentagons, and has $N = 42$. The number of geometrically distinct edges is again two, one corresponding to the edges of the icosidodecahedron and the other to the edges linking a pentagon vertex with the vertex at the pentagon's center. The exchange constants are again parametrized as $\cos\omega$ and $\sin\omega$ respectively. Figure 13 shows the reduced ground-state energy per bond of Hamiltonian (1) plotted as a function of $\omega$ for $d = 1$ to 5. At $\omega = 0$ the lowest-energy configuration of the icosidodecahedron is two-dimensional. Then for finite $\omega$ the MDGS develops in $d = 5$ dimensions, with the maximum energy per bond occurring at $\omega = 0.256725\pi$, being equal to -0.4403875 for both unique bonds (Table 2). This value is higher than the corresponding one for the pentakis dodecahedron. At $\omega = 0.34593\pi$ the MDGS becomes four-dimensional, and at $\omega = 0.41416\pi$ the spins become collinear.

Figure 14 shows the ground-state nearest-neighbor correlations as a function of $\omega$. Again there is a unique correlation for each unique exchange interaction in Hamiltonian (1), and the two correlations become equal at the point of maximum frustration. At the discontinuity the total spin jumps from zero to a finite value. The stronger frustration of the pentakis icosidodecahedron than the pentakis dodecahedron is also visible at the $\omega = 0$ limit, since it reduces to the icosidodecahedron and not to the dodecahedron, with the latter less strongly frustrated.

The pentakis snub dodecahedron has 72 vertices and four symmetrically independent edges and lacks a center of inversion. The MDGS achieves a three-dimensional structure in spin space (Table 2). Its maximum value equals $-\frac{\sqrt{5}}{5}$ and occurs when the only nonzero exchange interactions are between the 5-fold coordinated spins and their nearest neighbors, and

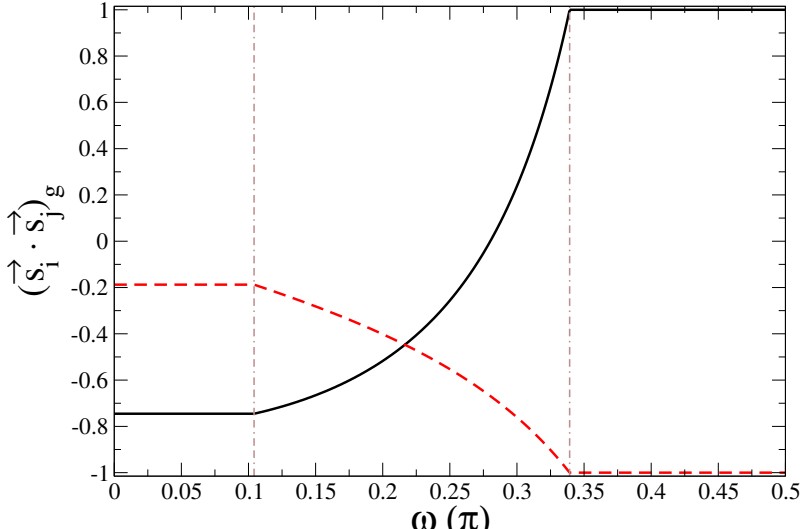

Figure 12: Ground-state nearest-neighbor correlations $(\vec{s}_i \cdot \vec{s}_j)_g$ for Hamiltonian (1) as a function of $\omega$ for the pentakis dodecahedron. The (black) solid line corresponds to the dodecahedron-edge correlation and the (red) dashed line to the pentagon vertex with the pentagon's center correlation. The corresponding interactions in Hamiltonian (1) are equal to $\cos\omega$ and $\sin\omega$ respectively. The vertical lines show the $\omega$ values where the dimensionality of the MDGS changes from 3 to 4 and then to 1.

among these nearest neighbors themselves, with the ratio of the two equal to $\frac{\sqrt{5}+1}{2}$. This is the minimally three-dimensional state that has the maximum ground-state energy for an isolated pentagon with a spin at its center. On the other hand the hexapentakis truncated icosahedron, which has 92 vertices and four symmetrically independent edges, has a five-dimensional MDGS in spin space. Its maximum ground state corresponds to a minimally three-dimensional configuration and develops exactly as the one of the pentakis snub dodecahedron.

The geodesic icosahedra have five and six-fold coordinated vertices. Each one of the former resides at the center of a pentagon, and forms with it a structure that achieves a maximum ground-state energy per bond of $-\frac{\sqrt{5}}{5}$, which equals the ground-state energy of the icosahedron [9, 28, 29]. The pentakis dodecahedron achieves this maximum in four spin dimensions, while the pentakis icosidodecahedron has a higher maximum possible energy than $-\frac{\sqrt{5}}{5}$ in five spin dimensions. On the other hand the hexapentakis truncated icosahedron achieves the maximum when the pentagons with a spin at their center are isolated from the rest of the cluster, which only requires three dimensions in spin space. What distinguishes this molecule from the two smaller ones is that the pentagons with a spin at their center are isolated from one another. The pentakis snub dodecahedron, which has $I$ symmetry and also isolated pentagons, maximizes the energy in exactly the same way, even though it has $n_g = 3$. These results show that frustration as determined from the maximum ground-state energy criterion does not necessarily weaken with the increase in the number of vertices, which makes the twelve five-fold coordinated vertices much less in number than the six-fold ones. The specific connectivity of each cluster is important, with the pentakis icosidodecahedron having the maximum possible ground-state energy per bond.

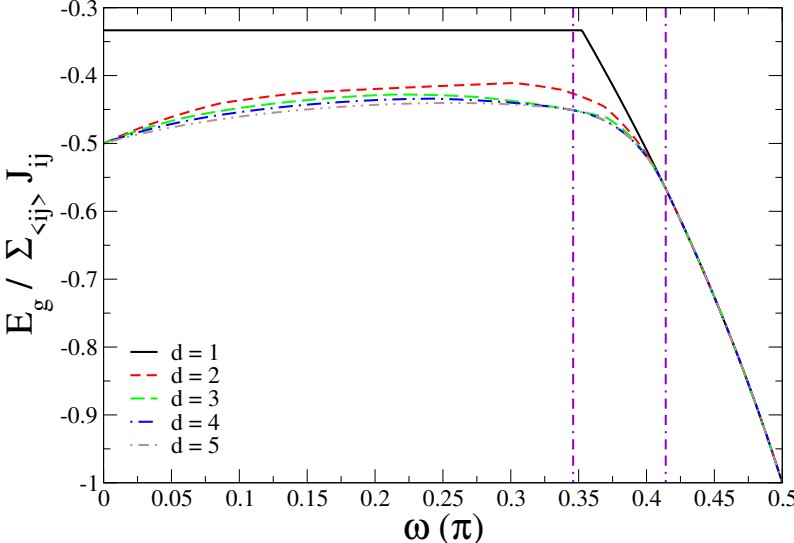

Figure 13: Reduced ground-state energy per bond $\frac{E_g}{\sum_{<ij>} J_{ij}}$ for Hamiltonian (1) as a function of $\omega$ for the pentakis icosidodecahedron. The icosidodecahedron-edge interaction equals $\cos\omega$ and the pentagon vertex with the pentagon's center interaction equals $\sin\omega$. The dimensionality of spin space $d$ ranges from 1 to 5. The vertical lines show the $\omega$ values where the dimensionality of the MDGS changes from 5 to 4 and then to 1.

## 9 Conclusions

Magnetic frustration has been characterized by the minimum dimensionality of the absolute ground state of the $n$-vector model, by allowing $n$ to take arbitrary values with the spins mounted on the vertices of different molecules being more than three-dimensional. Molecules of high symmetry such as Platonic solids in three and four dimensions (Table 1) and Archimedean solids (Table 2) have been found to form MDGSs in a number of spin-space dimensions equal to their real-space dimension. When there is more than one unique type of vertex, as in the case of fullerene molecules and geodesic icosahedra, the ground state can minimally develop in as many as five spin-space dimensions, in order to optimize nearest-neighbor interactions.

Frustration is also characterized by the maximum ground-state energy per bond when there are more than one symmetrically independent exchange interactions and they are allowed to vary. Typically the nearest-neighbor correlations are then equal, unless frustration does not exceed the one at the level of the maximally frustrated polygon of the molecule, as in the case of the $I_h$-symmetry fullerenes. This second way of characterizing frustration also demonstrates the existence of symmetry patterns within the same family of molecules. Furthermore, increasing the number of vertices does not necessarily weaken frustration as is expected, but the molecular point-group symmetry also plays an important role, as well as the specific connectivity of each molecule if they are of the same symmetry. It is also found that going from a dimension $n$ to the next higher one by switching on the interactions in the new dimension, it typically takes a finite strength of the latter to lower the ground-state energy.

The study of the $n$-vector model with arbitrary $n$ allows a more precise characterization of the frustration introduced by the molecular connectivity. This is because the energy minimization is not constrained by the requirement that the spins can only be up to three-dimensional,

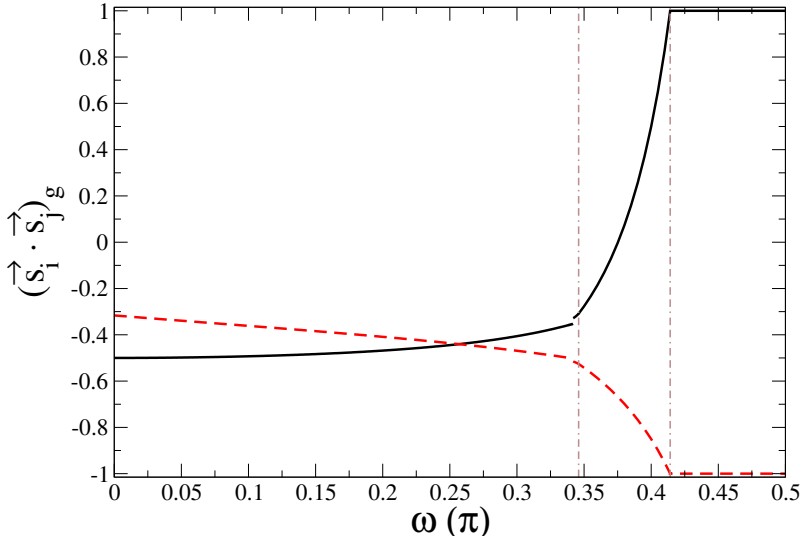

Figure 14: Ground-state nearest-neighbor correlations $(\vec{s}_i \cdot \vec{s}_j)_g$ for Hamiltonian (1) as a function of $\omega$ for the pentakis icosidodecahedron. The (black) solid line corresponds to the icosidodecahedron-edge correlation and the (red) dashed line to the pentagon vertex with the pentagon's center correlation. The corresponding interactions in Hamiltonian (1) are equal to $\cos\omega$ and $\sin\omega$ respectively. The vertical lines show the $\omega$ values where the dimensionality of the MDGS changes from 5 to 4 and then to 1.

allowing them to minimize the nearest-neighbor interactions more efficiently. The method introduced here can reveal more symmetry patterns if the computational resources are available to study molecules with more than four symmetrically independent exchange interactions.

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
