# Peer review of "Competition between frustration and spin dimensionality in the classical antifer romagnetic $n$-vector model with arbitrary $n$"

_SciPost Physics, doi:SciPost Phys. Core 6, 042 (2023)_

## Round 1 · Referee Report · Anonymous · 2022-8-30

Report
The present paper deals with ground state properties of certain classical spin systems with a large point symmetry including Platonic and Archimedean solids, fullerenes and geodesic icosahedra. The focus of the study is on two refined criteria for frustration that relate to the dimensionality of ground states and energy maxima with respect to different couplings. Concentrating on these aspects makes it possible to structure the abundance of material to some extent. In principle, the author has succeeded in this, with suggestions to further clarify the text following below.
A conceivable objection to the publication of the work would be the lack of reference to physical systems. Physical references are mainly made in the first and last sentence of the paper. However, they could be supplemented by references to the realization of some models by magnetic molecules, e.g.,
[a] A. Müller, S. Sarkar, S. Q. N. Shah, H. Bögge, M. Schmidtmann, S. Sarkar, P. Kögerler, B. Hauptfleisch, A. Trautwein, and V. Schünemann, Angew. Chem., Int. Ed. 38, 3238 (1999)
for the icosidodecahedron. Moreover, I think it is appropriate to consider a number of purely mathematical examples for the investigation of frustration criteria, independent of their physical or chemical realization. Therefore, in my opinion, the present material is valuable and definitely worthy of publication. A number of suggestions for revision and improvement of the paper follow.
1. The first sentence of the conclusion reads, "The magnetic frustration was characterized by the dimensionality of the absolute ground state of the n-vector model...". This suggests to the inexpert reader that there is always a single ground state of definite dimensionality. But, as the author explains elsewhere in the text, sometimes there are degenerate ground states of different dimensions. In this case obviously the minimal dimension is meant. If you limit yourself to the smallest dimension, the information for which dimensions real ground states exist is lost. For example, in the case of the Heisenberg icosidodecahedron, there are ground states of dimensions 2 and 4, but presumably none of dimension 3. It would be interesting to note whether something similar holds for the other cases considered in the paper.
2. Are there any particular reasons for the selection of examples in the paper?
3. The numerical procedure to find ground states is described as follows (at the end of section 2):
” A random initial configuration of the spins is selected and each angle is moved opposite its gradient direction, until the lowest energy configuration is reached. Repetition of this procedure for different initial configurations ensures that the global energy minimum is found.”
Just as a side note: it is clear that the global energy minimum can practically never be found with the mentioned method, but only approximated. This method is called "Iterative Minimization" (IM) in the literature, see [b,c,d]. Other methods, e.g., variants of the Monte Carlo simulation, have also been used to find ground states. For a detailed comparison of the different methods see [d], p.85 ff. It would be appropriate to say a few words about why the author prefers IM.
[b] L. R. Walker and R. E. Walstedt, “Computer model of metallic spin-glasses,” Phys.Rev. Lett., vol. 38, pp. 514–518, Feb 1977.
[c] S. R. Sklan and C. L. Henley, “Nonplanar ground states of frustrated antiferromagnets on an octahedral lattice,” Phys. Rev. B, vol. 88, p. 02440, Jul 2013.
[d] M. L. Baez, “Numerical methods for frustrated magnetism: From quantum to classical spin systems”, Dissertation, FU Berlin, 2018.
4. The citation policy of the paper is sometimes a bit strange. For example, 11 papers are cited [17-27] to indicate something that is not done in the paper. Similarly, the author refers to his own earlier work in the blanket form "The calculations were done numerically [15, 31-40]." Later, these works are definitely cited again in the appropriate places.
5. Figure 10 and Figure 11 show the same thing, only with different gradations of d. Could these figures be combined into one?
6. Figure 12: Which curve corresponds to which correlation?
7. The last sentence of the paper reads:
“Furthermore, the n-vector model can be taken as an effective model, for example in the case of spin-orbit interactions.”
Are there any references to this?
Minor point:
On p.5, 5th line from below, and at further places: “noncolinear”. The word “colinear” exists and means, e.g., in “colinear map”: the dual notion of a linear map. Here one should use ”collinear” meaning: lying on the same straight line.
Author: Nikolaos Konstantinidis on 2022-09-17 [id 2824]
(in reply to Report 2 on 2022-08-30)
I thank Referee 2 for the report. The following references have been introduced to make a connection with physical systems: For the icosidodecahedron: 10-14 on page 2. For Archimedean solids: 16 on page 2. For the truncated icosahedron: 17-22 (page 2). For the icosahedron: 26-27 (page 2). For the fullerene molecules: 43 (pages 3 and 12). For the dodecahedron: 48-50 (page 6).
With respect to the individual points raised by the Referee, the answers follow:
-
The answer to this question would require an extended search in the exchange parameter space of the molecules, associated with a careful analysis to identify the degenerate ground states in spin dimensions higher than the minimum ground-state dimension. A calculation of this type would have to take place for each molecule individually, even for the Platonic solids that only have a single geometrically unique exchange interaction, since the corresponding analysis needs to be meticulous. Such a calculation would require a significant amount of time and would be the object of a separate publication. According to the suggestion the expression "dimensionality of the absolute ground state" has been changed to "minimum dimensionality of the absolute ground state", with analogous modifications for similar expressions in the paper. These changes are among the ones that are highlighted with red color in the revised version of the paper.
-
The molecules selected in the paper have triangles and pentagons, which are the smallest frustrated polygons. The molecules selected have also been the object of study by me in the past, up to three spin-space dimensions, making them more familiar. The four-dimensional versions of the Platonic solids are considered for the first time, and are of interest especially since the spins are vectors that are allowed to exist in more than three spin-space dimensions.
-
To emphasize that the minimum is progressively approximated by the numerical method, I have added at the end of the last sentence of Sec. 2 with red color the phrase "within the numerical accuracy of the calculation". Reference [d] provides a very good overview of the different methods and have been added as a citation. The method used in this paper is the only one available, and it has provided accurate results in the papers by me referred throughout the paper.
-
The 11 references [31-41] (their numbers have changed with the addition of new references) are related to generalized models with spins in more than three dimensions. The references to the method have been removed from Sec. 2.
-
I prefer the figures to be different and not be combined in one. Introducing the details of Fig. 11 to Fig. 10 would make it look cluttered. Furthermore, if Fig. 11 is introduced in Fig. 10 as a small window, similar problems will occur, while the visibility of the data of Fig. 10 itself will be reduced.
-
The curves have been identified in the caption of Fig. 12, the same needed to be done for Fig. 14. The changes in the text of the captions are highlighted with red color.
-
No reference has been found. The sentence has been removed.
With respect to the minor point, the word collinear has been written with two l, and the changes are highlighted in red color.
I have also included in the text and highlighted with red color results on the pentakis snub dodecahedron, a geodesic icosahedron. The calculations were done whle the paper was under review.
Anonymous on 2022-09-27 [id 2854]
(in reply to Nikolaos Konstantinidis on 2022-09-17 [id 2824])
Answer to the author
It must be acknowledged that the author has taken up a number of criticisms and suggestions from the expert reports and that this work has been improved as a result. However, I miss an adequate consideration of two points from my report.
1) Talk of " the ground state" suggests a uniqueness that is often not given, even if one means (in the Heisenberg case) the equivalence class of spin configurations arising from rotations and reflections. In certain cases, this sloppy way of speaking, which ignores the degeneracy of ground states, may be tolerable; in the present work, however, it is misleading precisely because different dimensions of ground states are considered. Strictly speaking, the first sentence of the abstract “A new method to characterize the strength of magnetic frustration is proposed by calculating the minimum dimensionality of the absolute ground state of the classical nearest-neighbor antiferromagnetic n-vector model with arbitrary n.” makes no sense if one understands "the absolute ground state" literally. Similar examples run throughout the paper. In most cases, it would be sufficient to introduce the plural "ground states" to avoid a misleading term.
2) In my opinion, it is a good scientific standard to critically reflect on the central methods used and to cite predecessors who have also used these methods in a publication. Unfortunately, the author has not taken up my corresponding suggestions and references to the "Iterative Minimization", except for the citation [d], but even that not in the intended context. The answer of the author “The method used in this paper is the only one available, and it has provided accurate results in the papers by me referred throughout the paper.” seems to be somewhat unclear in the first part, while the second part of the statement could well be included in the paper.
I expect the author to consider these two points in a further revision or to present convincing arguments why he does not.
In my first report I was not aware of the possibility that the paper under consideration could also be published in SciPost Physics Core. When comparing the criteria for publication in SciPost Physics and
SciPost Physics Core I think that the criteria for the latter including
- Address an important (set of) problem(s) in the field using appropriate methods with an above-the-norm degree of originality,
- Detail one or more new research results significantly advancing current knowledge and understanding of the field,
fit the paper better than a qualification as "groundbreaking theoretical/experimental/computational discovery" or similar, as required for SciPost Physics.
Anonymous on 2022-10-26 [id 2956]
(in reply to Anonymous Comment on 2022-09-27 [id 2854])
-
The paper does not intend to study degenerate ground states existing in higher dimensions than the one of the minimum-dimensions ground state. This is a different project requiring separate attention. It is a common practice found in papers not to discuss something other than the minimum-dimension ground state, this does not mean that higher-dimensional ground states do not exist. Doing otherwise can easily lead to confusion, especially if the higher-dimensional ground states are implied in the text without being calculated. To minimize potential confusion I have introduced an acronym, MDGS (Minimum Dimensionality of the Ground States), explained in reference 10. I have included it throughout the text. Changes in the text are highlighted in blue (red color highlights changes after the first round of review).
-
In the first version of the paper, the phrase "Calculations were done numerically" was followed by a series of references were the method used in the paper was previously employed. Following comment 4 of the first report, I removed these references which were cited later in the paper. Now I introduce reference 25, that used the present method of minimization for the first time in 1992, reference 47 that used the method in 2007, and 48 that provides more information. Here the method is extended in arbitrary spin dimensions, as explained later in the text. The paper does not intend to provide details or compare minimization methods, as is typically also the case for similar papers. The use of the method has not been the result of a wider consideration and comparison of the available methods, but was a direction taken many years ago, as the dates of the references attest. This is what was meant by "the method is the only one available". Regarding the accurate results that have been produced in the past, they were given in the initial set of references that was removed after the Referee's comment in the previous round, but like the Referee pointed out these can be found throughout the text. I also included the two references that the Referee had suggested along with the one that is already in the paper, reference 51. First I describe how the calculation was exactly done, and then I cite the references suggested, in a more general context.
-
With respect to the appropriate journal for the article, it is the first time in the literature that frustration has been thoroughly examined in dimensions higher than three in spin space. To quote comment 4 of the Referee's first report, "For example, 11 papers are cited [17-27] to indicate something that is not done in the paper." Indeed such a calculation has never been done before, and it was not possible to the best of my knowledge to find directly related references. The paper provides physical insight on the manifestation of frustration in connection with symmetry, which is typically not revealed up to three spin-space dimensions. Also, the paper gives results on the ground state of molecules never given before and for the whole range of symmetry-allowed parameters, such as Archimedean solids (with ground states in three dimensions), fullerenes, geodesic icosahedra, and four-dimensional Platonic solids, even providing an analytic expression for the ground state of the latter. The calculations were possible through an extension of classical minimization in generalized coordinates in more than three dimensions and arbitrary dimensionality, which has not been done before. I believe that for these reasons the paper merits publication in SciPost.
Author: Nikolaos Konstantinidis on 2022-09-17 [id 2825]
(in reply to Report 3 on 2022-09-08)I thank Referee 3 for the report. With respect to his first point on the weaknesses of the paper, and according also to the suggestion of Referee 2, the following references have been introduced to make a connection with physical systems: For the icosidodecahedron: 10-14 on page 2. For Archimedean solids: 16 on page 2. For the truncated icosahedron: 17-22 (page 2). For the icosahedron: 26-27 (page 2). For the fullerene molecules: 43 (pages 3 and 12). For the dodecahedron: 48-50 (page 6).
With respect to the suggestions:
I have introduced two subsections in the Introduction.
The changes in the colored lines in the plots have been implemented.
If the plots as a function of d are extended farther than the ground-state dimensionality to clarify that the ground-state energy does not change any further, it is possible that the reader understands that the absolute minimum has not been reached yet. For example, Fig. 2 shows the ground-state energy as a function of d, and there are regions where the energy remains the same until it starts to decrease again. Allowing the plot to go beyond d=3 would create the impression that the energy will eventually decrease again, but this is not captured by the calculation. The way the graph is now makes it more transparent than above d=3 the energy will not change. It is also more transparent now that as d increases the dimensionality width for which the energy decreases is becoming narrower.
This phrase at the end of Sec. 2 has been changed to "generates the energy minimum" and is highlighted in red color.
The reason for not including the icosidodecahedron, the truncated tetrahedron, and the truncated icosahedron in Table 2 is that their frustration does not become stronger than their individual polygon units, and also that the information on their ground-state energy at least has already been published by other authors. The other molecules in Table 2, the truncated dodecahedron and the rhomboicosidodecahedron, whose frustration is not stronger than their polygon units, have not been presented in the literature before.
The change to n_g has been made throughout the paper and is highlighted in red color.
The word collinear has been written with two l, the changes are highlighted in red color.
With respect to the specific questions and comments:
It is true that the classification of frustration has to also take into account the arrangement of the triangles and the pentagons. The truncated tetrahedron is an Archimedean solid that includes triangles and not a fullerene. While the truncated icosahedron, which is a fullerene, also happens to be an Archimedean solid, Sec. 7 focuses on the fullerenes, which have as their basic frustrated unit the pentagon, as is also the case for the truncated icosahedron. Sec. 7 refers to fullerene molecules of T_d symmetry, that also have the pentagon as their frustrated unit, and compares the I_h fullerenes with them. Another argument pointing to weaker frustration for I_h fullerenes comes from the quantum model studied in Ref. 58 by Rausch et al., where the first excited state for uniform exchange interactions is a triplet and not a singlet.
In order to characterize frustration in the classical case, the exchange parameter space has to be investigated and also spins are allowed to have more than three components, which is done in the present paper. In Ref. 57 the low-energy spectrum is calculated for the Heisenberg model and for uniform exchange interactions. The comparison would be complete if the quantum calculation would have been extended to the n-vector model and also to any possible combination of the exchange parameters. In fact the results of Ref. 55 for the uniform exchange interaction case again do not provide generalized conclusions, and these were reasons while the study in the present paper was undertaken.
With respect to the very general questions and comments:
The distinction with respect to molecular real-space dimensionality has to do with the geometrical structure only. Furthermore one can characterize these molecules as zero dimensional, with two and three dimensions reserved for extended systems (square-triangular lattice, cubic lattice etc.). The characterization as three or four-dimensional in this paper is reserved in order to stress the difference between real-space dimensionality of the molecules and the spin-space dimensionality of the ground state, which can be more than three within the framework of the n-vector model. It is also kept in order to stress the inclusion of the four-dimensional molecules in the paper.
In Table 1, which lists molecules with all edges and correspondingly exchange interactions equivalent, this parameter is given in the last column, and it is the ground-state energy per bond, which corresponds to the nearest-neighbor correlation and angle for any pair. Comparing the molecules that have the same fundamental frustrated unit shows that as the number of spatial dimensions increases the energy per bond increases. In Table 2 molecules with more than one unique exchange interaction are listed. The corresponding column is the one that lists the maximum possible ground-state energy, which frequently coincides with all the nearest-neighbor correlations. For quite a few of the molecules this value is equal to a value from Table 1, showing that frustration is weaker (for the I_h symmetry molecules for example), while for others this value is higher, indicating stronger frustration. The ground-state energy per bond (or average nearest-neighbor correlation and angle) is also plotted in various graphs in the paper.
The geodesic icosahedra are derived from the icosahedron by subdividing each face into smaller faces using a triangular grid (Ref. 45). References for the icosahedron and its realization are given in the paper, but the geodesic icosahedra do not exist as molecules yet. I have worked on the icosahedron (references given in the paper) and the first derived geodesic icosahedron, the pentakis dodecahedron (Ref. 59), and wanted to extend the study to bigger molecules of the family, since they are also the duals of the I_h fullerenes and therefore highly symmetric. The similar properties of the dodecahedron and its dual, the icosahedron, have been shown in Ref. 56.
I have also included in the text and highlighted with red color results on the pentakis snub dodecahedron, a geodesic icosahedron. The calculations were done whle the paper was under review.

---

## Round 1 · Referee Report · Anonymous · 2022-9-8

Strengths
1. contributes to the task of providing a definition or measure for frustration
2. brings some classification into the zoo of molecules
Weaknesses
1. rather narrowly focused and needs to be better embedded into the state of research
2. readability of plots needs to be improved
Report
The paper aims to quantify the strength of frustration by using the n-vector classical spin model, where additional spin components are ramped up, allowing one to smoothly vary the dimensionality in spin space. The aim is to see at what dimensionality the energy reaches its absolute minimum - the higher it is than the real-space dimensionality of the system, the stronger the frustration. This is applied to various molecules/clusters.
I find the idea interesting, since it (1) aims to provide a definition/measure for frustration, which is surprisingly difficult to do; and (2) brings some classification into the zoo of molecules. However, I think the paper leaves a lot of questions open and requires some more improvements.
Requested changes
Some relatively trivial suggestions:
- The introduction contains a large chunk of text with no subdivisions. A further structuring using subsections would be nice for the reader.
- All colored lines in the plots need to be thicker for better readability.
- The plots as a function of d generally end at the ground-state dimensionality, but should be extended a bit further, so that it becomes clear that this is indeed the absolute minimum with respect to varying d.
- The formulation "ensures the global energy minimum" should be probably softened. I think the only way one can ensure to have found the global minimum is by plotting the full energy landscape, otherwise the danger of a local minimum always exists. But this is a generic problem, of course.
- It would be nice to have the Archimedean solids mentioned in the introduction (icosidodecahedron, truncated tetrahedron, truncated icosahedron) to also appear in Tab. 2 for the sake of completeness.
- The ground-state dimensionality "n" in the tables should be something like "n_g", similar to "E_g", since n is a running variable.
- It think the more common spelling is "collinear" rather than "colinear".
Some specific questions and comments:
- In Sec. 7 it is claimed that the strength of frustration is related to the symmetry, so that I_h tends to be less frustrated, while T_d tends to be more frustrated. But it seems that this rather has to do with the arrangement of the odd-valued faces (which the author also mentions). For example, the truncated tetrahedron is T_d, but has isolated triangles and it is mentioned in the text that it has n_g=3. On the other hand, I expect that an I_h fullerene with clustered pentagons will be strongly frustrated, though perhaps it's not possible to find one.
- It seems to me that this characterization of frustration doesn't match up well with the quantum model, e.g. with the characterization via low-lying singlets. From PRB 80, 134427 I gather that C24 has 1 singlet, C28 has no singlets, C30 has 3 singlets before the first triplet. Yet, C28 has n_g=4 and would have the strongest frustration of the three in the classical case. I wonder how it is for C36, which the author has diagonalized in JMMM 449 (but where the low-lying singlets are not given)? Can the author comment on the relationship between the classical case and the quantum case?
Finally, some very general questions and comments:
- The molecules considered have the topology of a sphere and are locally 2D, albeit with a curvature. Is it justified to see them as 3D objects? There are other molecules where the inside of the sphere is filled as well (see e.g. Kaatz and Bultheel, 2019), which seem more qualified to be called 3-dimensional.
- The classification via the dimension n_g is very discrete and many objects land in the same category. Maybe it would be useful to have an additional continuous parameter, e.g. an angle deviation from collinearity, which is mentioned in Sec. 3, but not carried through.
- I'm curious about the choice of geodesic polyhedra. Do they exist as molecules? If so, a reference would be nice. Otherwise, I wonder why they are chosen over the many other magnetic nanoclusters that are in existence.

---

## Round 2 · Referee Report · Anonymous (Referee 3) · 2022-9-20

Report
According to the abstract, the paper proposes a "new method to characterize the strength of magnetic frustration". I understand this as a broad statement about the geometry that should have implications for both the quantum and the classical case. The more narrow scope would be an investigation of the n-vector model on various frustrated geometries.
It seems to me that the author is inconsistent on this point: In the reply, he cites the excited states of C60, which corroborate his approach; but dodges the question on the mismatch for other fullerenes as "incomplete comparison" because the quantum case has only three spin components.
If the author makes an earnest effort to discuss the implications of his frustration measure for the quantum case (where it works and where it doesn't, how it compares to other measures etc.), I can accept the premise of the abstract and recommend it for a publication in SciPost. If the author wants to make a more narrow statement on the n-vector model, I would recommend a publication in SciPost Core.
Since no one has come up with a perfect measure of frustration yet, I realize that it makes no sense to be very strict here; but I think that at least an effort should be made.
It seems to me that the author is inconsistent on this point: In the reply, he cites the excited states of C60, which corroborate his approach; but dodges the question on the mismatch for other fullerenes as "incomplete comparison" because the quantum case has only three spin components.
If the author makes an earnest effort to discuss the implications of his frustration measure for the quantum case (where it works and where it doesn't, how it compares to other measures etc.), I can accept the premise of the abstract and recommend it for a publication in SciPost. If the author wants to make a more narrow statement on the n-vector model, I would recommend a publication in SciPost Core.
Since no one has come up with a perfect measure of frustration yet, I realize that it makes no sense to be very strict here; but I think that at least an effort should be made.
Requested changes
- a discussion of the implications of the proposed frustration measure for the quantum case
- I think an inclusion of the icosidodecahedron, the truncated tetrahedron, and the truncated icosahedron in Tab. 2 is still helpful to have a comprehensive list (even if the results have been published before), but this is up to the author

Author: Nikolaos Konstantinidis on 2022-10-26 [id 2955]
(in reply to Report 1 on 2022-09-20)In order to perform the calculations in the paper, a method of classical minimization had to be used in more that three spin-space dimensions for the first time, and the method can be used up to arbitrary spin dimensions, limited only by computational resources. Consequently the calculations were very time consuming, especially since there is typically a number of symmetry-independent unique exchange interactions. Still, a relatively large number of molecules has been considered (to quote one of the remarks of the Referee from the first round of reviewing: "brings some classification into the zoo of molecules".)
To set up the analogous quantum-mechanical calculation would be very tedious, since one has to do it in more than three spin-space dimensions, and this calculation would have to be set up. To make matters worse, even to examine a small fraction of the molecules considered in the paper would take a very long time, since the quantum-mechanical calculations are significantly more time consuming. It must also be remembered that, unlike classical calculations, the calculation of the lowest-energy state for low-energy sectors of large molecules is not possible. In fact, calculating the low-energy spectrum of C60, reference 62 in the text, requires the use of DMRG and ended up as a separate publication in the current journal, SciPost. The corresponding quantum-mechanical project would be very time consuming, even in comparison with the current classical one.
With respect to the reference to C60 (reference 62), according to the text fullerenes of its symmetry have classical three-dimensional ground states, so a comparison with the quantum states (of three dimensions at least) is possible. This comparison is far from complete, but having the quantum-mechanical low-lying states for C60 is very challenging even for equal exchange interactions (and this is a paper published in SciPost). On the other hand, molecules like the fullerene with 28 vertices have more than three-dimensional classical ground states, and a comparison with the corresponding quantum states of reference 61 is not possible. This leaves as the only possible candidates with three-dimensional classical states the 24 and 30 sites fullerenes, since the other molecules with three-dimensional ground states are too big. As was stated in the previous paragraph, a formal calculation would first treat the quantum mechanical problem in arbitrary dimensions, which would be quite tedious. Even in three dimensions the time required would be significant. In reference 61 the quantum model was considered only for spatially uniform exchange interactions, unlike the classical case presented here, and those calculatons required significant time. It was also not possible to find the excited states of the 36 cite cluster in reference 59, due to computational limitations that still exist. Finding only the lowest-energy state in a specific irreducible representation of a specific Sz sector is technically easier than calculating more low-lying states in the same sector.
To summarize, it is highly challenging and not only time-wise to scrutinize the current method for the quantum case, to compare it with other measures etc. Only the corresponding calculations for the systems in this paper and only up to three dimensions, classical or quantum, have spanned many papers. Therefore I do not understand the purpose served by words such as "dodges" and "earnest".
With respect to the second requested change, the calculations in this paper have shown that for the truncated tetrahedron and the truncated icosahedron the lowest-energy classical ground states are three-dimensional and have been already found in the literature, so there is nothing new to be added. For the icosidodecahedron it was already known that the lowest-possible classical ground state is two-dimensional. Apart from the molecules associated with new results, I prefer to only include the odd polygons in the Tables since they are fundamental to the concept of frustration.
Anonymous on 2022-11-10 [id 3003]
(in reply to Nikolaos Konstantinidis on 2022-10-26 [id 2955])This is clearly a misunderstanding, I don't expect the author to make any new calculations that are beyond the scope of the paper, but merely to compare with existing data.
Let me try to reformulate: One common measure of frustration is given by the degeneracy of the classical Heisenberg model on a particular geometry. For example, the kagome lattice is infinitely degenerate, while the triangular lattice is not, and is in this sense stronger frustrated. This is a statement on the geometry. The strong frustration that manifests itself in the degenerate classical state will manifest itself differently in the quantum case; but generally, we can use the classical result as a guide to identify interesting geometries and parameter regimes to look at in the quantum case.
As I initially understood the paper, it introduces another measure of frustration given by the dimension n_g. So if we have n_g=4 for C28, it similarly seems to imply that this particular geometry is stronger frustrated than others, which should manifest itself in some manner in the quantum case in the more physical 3-dimensional case.
If it does, then we have very interesting implications. For example, it would reveal the most interesting molecular geometries (such as the geodesic polyhedra with n_g=5) out of a large zoo of possibilities, where one would expect interesting states for other models as well. It would furthermore motivate to look for these specific molecules and states in the experiment.
(Please note that I'm trying to help to improve the paper.)
But such implications are not discussed and the author seems to maintain that the n-vector model can only be compared to the n-dimensional Heisenberg model. In this case, the scope of the paper is more narrow than I understood initially and I would tend to recommend a publication in SciPost Core.

---

## Round 2 · List of Changes

Changes are highlighted in red.

---

## Round 3 · Referee Report · Anonymous (Referee 4) · 2022-11-6

Strengths

It is a comprehensive work on magnetic frustration for various theoretical model spin systems.

Weaknesses

This work extends the existing knowledge about magnetic frustration in model spin systems, but I don't see any significant impact for that on the physics of these systems. Furthermore, I am missing a connection to existing systems and/or experimental studies.

Report

The manuscript does not meet the acceptance criteria of the journal. I recommend publishing it in SciPost Physics Core.

---

## Editorial Decision

published